# Description of Telemark Skiing Technique Using Full Body Inertial Measurement Unit

**DOI:** 10.3390/s23073448

**Published:** 2023-03-25

**Authors:** Piotr Aschenbrenner, Bartosz Krawczyński, Marcin Krawczyński, Tomasz Grzywacz, Włodzimierz Erdmann

**Affiliations:** 1Faculty of Physical Education, Gdansk University of Physical Education and Sport, 80-336 Gdansk, Poland; 2Department of Sport, Institute of Physical Education, Kazimierz Wielki University, Chodkiewicza 30, 85-064 Bydgoszcz, Poland

**Keywords:** 3D motion registration, analysis of technique, Telemark skiing, inertial sensors

## Abstract

Researchers involved in skiing investigations postulate Telemark skiing as an alternative technique to Alpine skiing, which may be associated with lower injury risk. A free heel of the boot, and a boot that enables flexion of the toe, are characteristic features. The aim of this research was to compare three types of turns on Telemark skis, through a biomechanical description of each skiing technique. Seven professional skiers were investigated. Two cameras and the MyoMotion Research Pro system were utilized. Eighteen wireless IMU sensors were mounted on each skier’s body. For every skier, five runs were recorded for each of the three turning techniques. Velocity of run, range of movement, angular velocity in joints, time sequences, and order of initialization of movement were obtained. A higher velocity of skiing was obtained during the parallel (14.2 m/s) and rotational turns (14.9 m/s), compared to a low–high turn (8.9 m/s). A comparison of knee angles, revealed similar minimum (18 and 16 degrees) and maximum (143 and 147 degrees) values achieved during the parallel and rotational techniques, which differed considerably from the low–high technique (27 and 121 degrees, respectively). There were no significant differences in trunk rotation angles. A detailed analysis of the Telemark skiing technique revealed novel information on how turns are executed by well-trained skiers and the impact of different approaches.

## 1. Introduction

The most crucial consideration in skiing technique has always been movement optimization, to increase performance while lowering the danger of injury. Currently, the Alpine skiing technique dominates, about 65% of all users on the slope are alpine skiers, 30% snowboarders, 2% Telemark and ski touring [1].

Many authors have emphasized that skiing is a sport that leads to frequent injury [1,2]. Some researchers, e.g., Federiuk et al. [3,4], have postulated that Telemark skiing offers an alternative approach to Alpine skiing, for lowering injury risk. Telemark skiing is characterized by the use of special equipment: skis with Telemark bindings, shoes, and sticks. A free heel of the boot (not fixed with the binding as in Alpine skiing) and a boot that enables flexion of the toe, are features. This promotes the Telemark position, namely forward–backward shifting of the feet, with a simultaneous flexion of the knee joint, particularly during turning movements.

In skiing, speeds of up to 16.5–20 m/s (about 60–70 km/h) are achieved in recreational skiing, and above 33 m/s (approximately 120 km/h) in competitive skiing. The mean radius of the curve is compatible with the design turning radius of the ski, which is about 11–16 m. Forces acting on a recreational skier’s body should not exceed twice the force of gravity, with extreme movements possibly being much higher. During the run, there is a full range of motion in the knee and the hip joints and, to a lesser extent, in the shoulder and the elbow joints [5].

The biomechanical aspects of ski turns have been widely studied, to understand the fundamental nature of the turning motion, with several devices and methods. Historically, the most common method was 2D [6] and 3D video analysis [7,8,9]. Technology developments have led to an expansion of methods for assessing skiing technique, including force plates and platforms [10,11,12,13], RTK GPS receivers [14,15,16], inertial sensors [15,17], and inertial measurement units (IMU) [18]. The application of IMU sensors offers several benefits for testing skiers in their natural environment, due to low invasiveness (they do not touch the skin and do not interfere with the movement) and high accuracy. Unlike other methods, the IMU approach is not limited by the measuring space; there is no need to set up equipment on slopes or perform time-consuming calibrations. For investigators, this also means less time in preparing the research area and it reduces the time burden on research subjects. Hence, Yu et al. [18] recommended this method to improve performance, prevent injuries, and design strategies in working with a ski team, before the 2018 Winter Olympic Games.

The aim of the study was to compare the Telemark skiing technique, among well-trained skiers, to the parallel and rotational techniques, which are better for novice skiers, as they produce lower centrifugal force and unnecessary leaning of the body. This was achieved by recording the range of motion, and time characteristics in joints, and specifying the order of initiating movement in particular body parts, during downhill runs on the same slope. It was hypothesized that the low–high technique in Telemark skiing, yields significantly different performance outcomes, compared to the parallel and rotational techniques. Aschenbrenner and Erdmann [19] have previously presented the research concept and preliminary research results. They demonstrated that detailed knowledge of these biomechanical differences provides a quantitative framework to identify, and thus correct, less efficient and/or more injury-inducing skiing techniques.

## 2. Materials and Methods

### 2.1. Study Design

The study design was a case series model, where velocity of run, range of movement, angular velocity in joints, time sequences, and order of initialization of movement were obtained. For every skier, five runs were recorded for each of the three turning techniques. The following three turning techniques, started by kneeling with one leg for unloading the body, were chosen for assessment: (1) low–high (for beginners), (2) parallel (for intermediate), and (3) rotational (for advanced).

### 2.2. Participants

Seven professional skiers (six males, one female), who are instructors and members of the national DEMO TEAM in Telemark, were tested. Their mean age was 36 years (standard deviation: SD = 8.2), height 168 cm (SD = 12.5), and body mass 72 kg (SD = 8.9). All of them had been involved in Telemark skiing for over ten years, and presumably, their technique was of the highest standard. The small number of participants was unavoidable, due to the limited number of skiers selected for this competition. All members of the national team were tested. The research was conducted during the grouping of the national team in Telemark skiing, with the consent and knowledge of the Association of Ski Instructors and Trainers, an institution responsible for ski training and supervision of safety during skiing. The only qualifying criterion, was membership in the team. Due to the small sample size, no participants were excluded. The study was conducted with the Declaration of Helsinki and approved by the Bioethics Commission of the Regional Physician Chamber in Gdansk, Poland (No. KB—22/17, 18 September 2017). All participants were informed about the research procedure, and use of results, before giving their informed consent to participate in this study. The tests were typical (non-invasive) activities that the subjects were familiar with, as part of normal daily training. No aspects of this research were considered detrimental to their health, as elite athletes.

### 2.3. Methods

To record skiers’ movement, the MyoMotion Research Pro system was utilized (Noraxon, Scottsdale, AZ, USA). Eighteen wireless inertial sensors were mounted on each skier’s body parts, according to the manufacturer’s recommendations and the experiences of Yu et al. [18] (Figure 1). Every sensor combines an accelerometer, a gyroscope, and a magnetometer, to measure the 3D turning angles of each sensor in absolute space (the so-called reference or navigation angles), on the basis of the so-called fusion algorithms. Subsequently, the sensors enable precise registration of the position of all body parts in space, and their angular velocity and acceleration. The Noraxon technology mathematically combines and filters out three of nine input sources at the sensor level, and transmits four quaternions from each sensor and optional information about 3D linear acceleration. The measurement accuracy of the sensors, based on the manufacturer’s specifications, was ±0.4° in static measurements and ±1.2° in dynamic ones. The sensors were fixed with elastic bands placed directly on the skin, or on thin underwear, but with sufficient stiffness to prevent the movement of displacement. Data from the sensors were recorded by a wireless data logger. The values of angles in the joints in the physiological planes of motion for a given joint, were used for the analysis. These values were calculated by the device software, using data from the sensors. The axes of the joints were determined during calibration in a standing position. The definitions of the planes were related to the long axis of particular body parts and the anatomical structure of the joints included in the model used in the software.

This research was conducted at an indoor ski slope (Snow Arena in Druskininkai, Lithuania). The venue is 460 m long and 50 m wide, with an even slope, across a height difference of 66 m. The skiers were instructed to move in curves, on a track marked with poles. The length of each section and the turning radius were measured. The skier’s movement was recorded using three video cameras. The first fixed camera (Sony HDR-PJ810, with a frequency of 200 fps) was placed in the middle of the turn and panned the movement of the skier. On the basis of the run time (counted frames) and the given distance between the gates, the average velocity on a given section was calculated. The second fixed camera was placed at the top of the slope. The third was a moving camera, which was held by a skiing operator that followed each test subject. For every skier, five runs were recorded for each of the three different turning techniques (15 downhill trials in total).

The following three turning techniques, started by kneeling with one leg for unloading the body, were chosen for assessment: (1) low–high (for beginners), i.e., just before a turn, lowering of the body occurs for unloading of skis, turning the skis outside the turn, and moving upwards; (2) parallel (for intermediate), i.e., by holding the skis parallel (about 10–15 cm apart), body weight is moved a little forward and towards the outside ski, this allows easier movement of the tail of the skis and sliding down with both skis; (3) rotational (for advanced), i.e., inside rotation of shoulders occurs, followed by inside rotation of the hips. These techniques form part of the education curriculum and examination procedures for skiing instructors, as introduced by the Polish Skiing Union. Skiing velocity in each downhill trial was adjusted to the type of turn and terrain, being a natural consequence for a given skier (expert opinion).

The range of motion in individual joints, the time and order of movement initiation, and the angular velocities in the joints were obtained, using a system for recording the position of body parts and video analysis. Based on this data, one can create a standard technique. Using a given velocity and the turning radius, the value of the centrifugal force, F, and the ground reaction force, as well as the required angle of leaning the body inwards (to balance the centrifugal force), were calculated.

The study data were subjected to standard statistical analyses, using the Statistica software. Every participant completed 15 trials (3 techniques × 5 runs). Repeatability was high and the technique presented was similar, so distribution was low. However, due to the small size of the group tested, the Lilliefors t-test and the Kolmogorov–Smirnov test both revealed a non-normal distribution (*p* < 0.05). The arithmetic mean, standard deviation (SD), median, range of quartiles, minimum (min), and maximum (max), were calculated for each skiing technique. Non-parametric (Friedman’s) tests were used to compare biomechanical differences between these techniques. If a statistically significant result emerged, a follow-up (Wilcoxon test) procedure was employed, to identify the location of the mean differences.

## 3. Results

The school turn “low–high” technique produced the slowest runs, with an average velocity of 8.9 m/s (SD = 1.8 m/s). Using the parallel and the rotational turns, a higher velocity, of over 14 m/s, and an increased turning radius, of 13.4 m and 14.5 m, compared to 8.9 m for the low–high turn, were achieved (Table 1). The kinematic values for the LH school turn were significantly different from the means of the other two methods

The IMU data signals were used to calculate the range of motion in specific joints over time (Figure 2). Table 2 displays these statistics. There were significant differences between the school turn and the final technique, namely the parallel and rotational turn. No significant differences between the parallel turn and the rotational one emerged.

In addition, the time of initiating movements in selected parts of the body was determined. The primary distinction between a parallel and rotational turn is the beginning of the motion; in a parallel turn, the shoulders, hips, and knees should rotate equally. Analyzing the change in the angle values over time, it can be seen that in parallel torsion, all the main parts of the body (i.e., the shoulders, pelvis, and knees) begin to move simultaneously, while in a rotational turn, the shoulder moves an average of 0.2 s before the hip (see Figure 3b).

## 4. Discussion

From the results of the IMU sensors, it can be quantitatively proved that a given movement was initiated by rotation of a hip, rather than a shoulder. These are the fundamental data for an analysis of a skiing techniques, relevant for a coach to assess the correctness of movement. The investigators proved the hypothesis that the low–high technique gave significantly lower velocity of the turn, which is safer for beginners. They also observed that the parallel turn was in coincidence with the rules of the union, but the rotational turn was not. In the latter, instead of the shoulder, the hip was moved first. The investigators stated that the IMU was a good equipment for obtaining technical data of the skiers. Up to now, visual assessment could not give such detailed information.

Pasek [20] paid attention to the need for outdoor physical activity. One of the ways of being active outdoors, is skiing of different forms, e.g., Alpine or Telemark skiing. The obtained velocities of Telemark skiing in this research, were similar to Alpine skiing velocity of recreational skiers. Shealy [21] gave a velocity of an average recreational skier of 12 ± 3.1 m/s (43 ± 11.2 km/h), on slopes with an inclination of 16–20°. Using the 3D position determination system, an accurate record of body movement over time was obtained and the directions and order of movement of individual parts of the body were determined. The IMU system has been used previously to study motion. Kang and Gross [22] identified the accuracy of inertial sensors in comparison to the optical system as very high—the intraclass correlation coefficients were 0.9. Brodie et al. [17] showed, in a laboratory test, an error in the IMU sensors placement during movement at the level of 0.8–1.3 degrees. Brodie et al. [14], in their study of the New Zealand ski team, applied the fusion motion capture (FMC) to capture the 3D kinetics and kinematics of Alpine ski racing. The system consisted of inertial measurement units (IMU), a global positioning system (GPS), pressure sensitive insoles, video, and theodolite measurements. Brodie used data from FMC to estimate the forces acting on a skier, energy, and power. A similar experiment was conducted by Supej [15], which also confirms the high accuracy of data obtained from inertial sensors.

In Telemark skiing, there is a mechanism of movement similar to the natural movements of walking. The skier enters the turn by changing and moving the leg forward and turning the upper body. Compared to Alpine skiing, where the forces are transferred mainly in the transverse plane by loading the knee and hip joints [7,11,15], Telemark skiing deals with loads in the sagittal plane, which is normal physiological work for the knees.

Only the angles in the lower limbs, trunk, and pelvis were considered for analysis in this study, despite measurements of the whole body. The position of the arms in the instructor’s demonstration should be constant, regardless of the turning technique. Differences between the tested turning techniques should be only in the described angles. The people surveyed were able to demonstrate this. No statistically significant differences were found in the analysis of upper limb angles. However, such differences may occur at the amateur level, and should be taken into account there.

Ways of initiating a turn, by creating an appropriate turning impulse, depend on the skier’s technical skills and motor abilities. To change the position of skis on snow, a skier act on them with their muscles. Most often, these are twists heavily overloading the joints of the lower limbs. Unloading the skis will facilitate moving them. In Telemark skiing, unloading is achieved naturally while changing the forward leg. While moving upwards, at the highest point, unloading takes place, which makes it easy to change skis (Figure 4). Because of this mechanism, Telemark turns have less torsion load on the lower limbs than Alpine skiing.

The Telemark turn technique has been previously studied. Tant et al. [23] provide a detailed description of how to execute a Telemark turn. This technique consists of three phases: the preparatory phase, force drop phase, and force steering phase. These phases were also observed in our movement analysis. Nilsson and Haugen [24] also describe the different phases of the turn. They investigated the activation of knee extensor muscles and knee angular displacement during competitive Telemark skiing. The specificity was evaluated based on angular amplitude, angular velocity, muscle action, and electromyographic (EMG) activity. Five expert male Telemark skiers participated in the study, skiing on Telemark skis and performing specific dry-land strength training exercises, such as Telemark jumps and barbell squats. The results showed that the knee angular displacement during Telemark skiing and dry-land Telemark jumps, had four distinct phases: a flexion and extension phase during the thrust phase of the outside ski/leg in the turn/jump, and the same when the leg was on the inside of the turn/jump.

Advanced measurement techniques such as IMU sensors, are commonly used in modern sports research [25]. Norimitsu et al. [26] have compared the Telemark turn and Alpine (parallel) turn, by visualizing graphs of strain distribution on the Telemark ski during turning. They measured the distribution of strain on the Telemark ski during a Telemark turn and an Alpine (parallel) turn, on a 90-meter and 11-degree compacted snow slope, with four flags, at intervals of 20 meters. The graphs of strain distribution showed that during the Telemark turn bending occurred on the backside ski during turning. This finding is consistent with our claim of less torsional load on the knees.

When analyzing movement techniques, one must not overlook the motor and mental limitations that can impact the quality of movement. The impact of these factors on skiing technique was studied by Makowski et al. [27]. They found that raising awareness among skiing instruction participants can increase their mental resilience and improve the final outcome of the instruction.

Accurate knowledge of body movements gives us many opportunities, e.g., to correct performance errors, determine dangerous overloads, and for athlete assessments by instructors. In targeting highly trained skiers, the results may not be replicated in lesser-trained or amateur skiers, under the same conditions. The methodology employed is also limited to a well-equipped research setting, due to the availability and cost of the equipment used. Future work could aim to create a lower-cost alternative to the current sensing and design software suite, perhaps integrated with mobile phone technologies, to broaden the scope of this work to a real-world setting.

## 5. Conclusions

The measurements obtained enabled an objective description of the body motion during Telemark skiing, using a range of kinematic parameters. It has been shown that, owing to the use of IMU sensors, it is possible to test this skiing activity in real conditions, without interfering with the tested person. With this knowledge, we are able to better assess and improve the efficiency of movement for Telemark skiers.

## Figures and Tables

**Figure 1 sensors-23-03448-f001:**
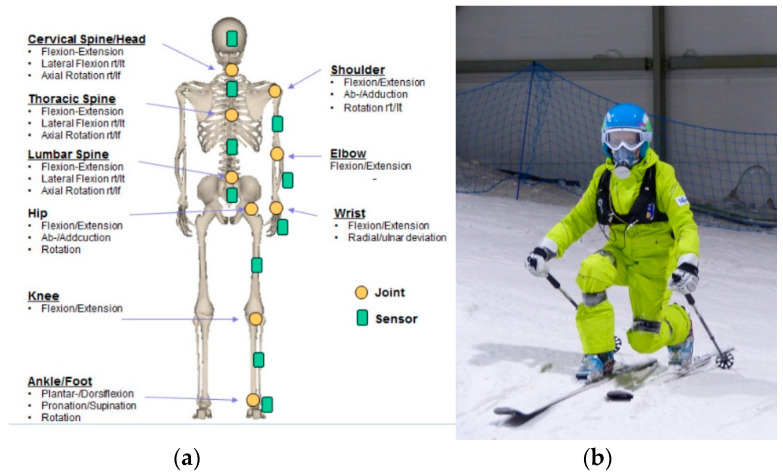
Inertial sensors: (**a**) placement on a skier (as per manufacturer’s instructions); (**b**) a skier with mounted sensors during the tests.

**Figure 2 sensors-23-03448-f002:**
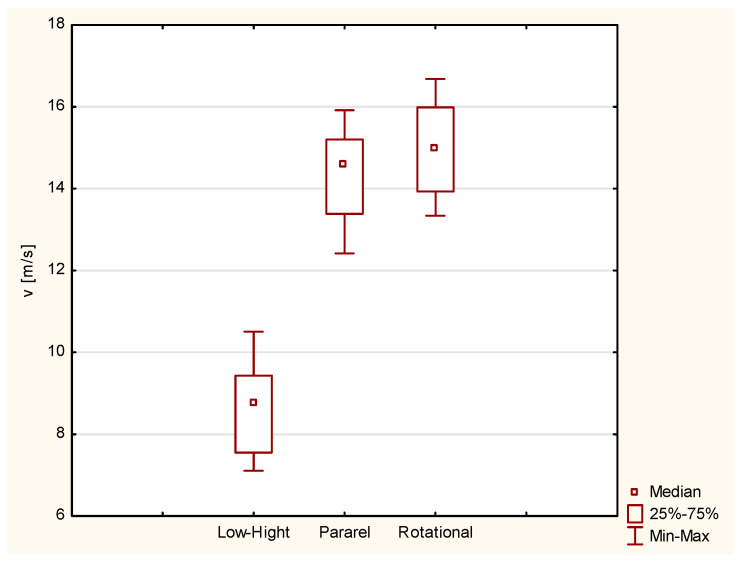
Velocity in the particular turns [m/s].

**Figure 3 sensors-23-03448-f003:**
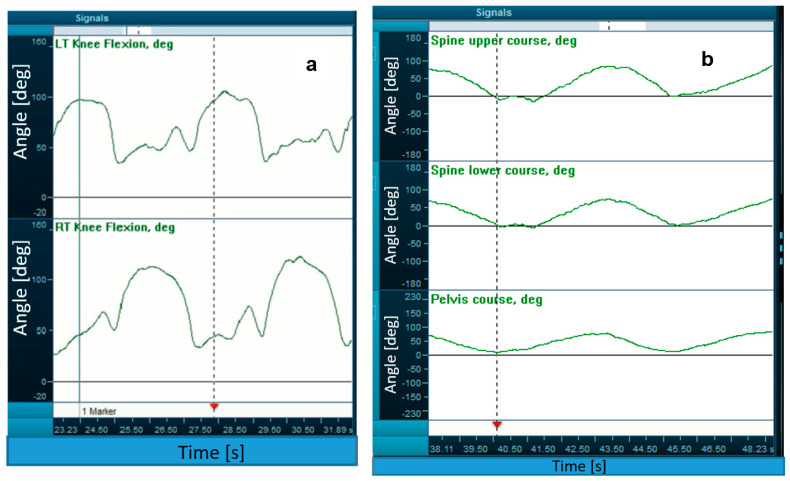
Technique characteristics: (**a**) range of motion in the knee joints; (**b**) time sequences in the rotational turn.

**Figure 4 sensors-23-03448-f004:**
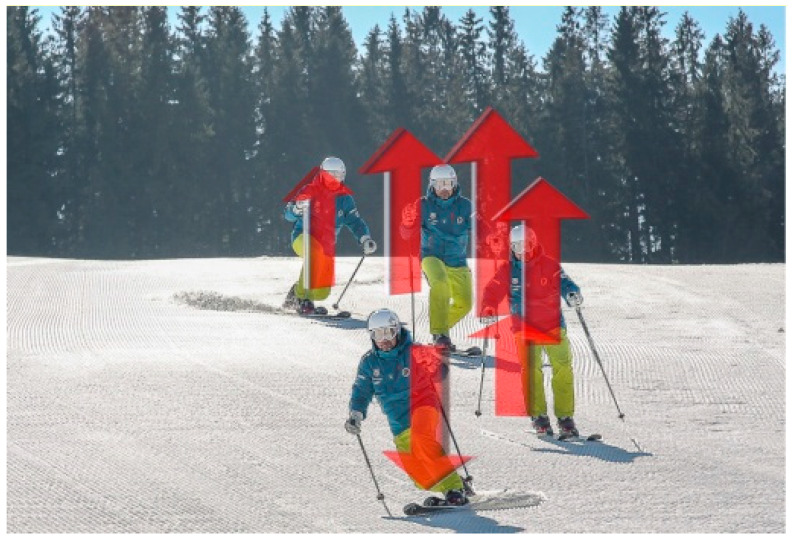
Unloading the skis in Telemark turn (the arrows show the vertical movement of the body).

**Table 1 sensors-23-03448-t001:** Basic kinematic data of the particular turns (mean ± SD).

	Velocity [m/s]	Radius [m]	Frequency of Turns [Hz]	Time of One Cycle [s]
Low–high	8.9 ± 1.8 *	9.2 ± 1.8 *	0.40 ± 0.2	5.02 ± 0.9 *
Parallel	14.2 ± 2.1	13.4 ± 1.8	0.43 ± 0.3	4.66 ± 0.7
Rotational	14.9 ± 1.9	14.5 ± 1.8	0.42 ± 0.3	4.68 ± 0.8

* a statistically significant different result.

**Table 2 sensors-23-03448-t002:** The kinematic values for motion.

		In the Knee Joint		
	Min angle [°]	Max angle [°]	Time of flexion [s]	Time of extension [s]
Low–high	27 ± 5.2 *	121 ± 6.8 *	1.5 ± 0.2 *	2.2 ± 0.5 *
Parallel	18 ± 2.2	143 ± 20.2	0.92 ± 0.1	1.36 ± 0.3
Rotational	16 ± 3.4	147 ± 11.8	0.93 ± 0.2	1.36 ± 0.2
		**In the Hip Joint**		
	Min angle [°]	Max angle [°]	Time of flexion [s]	Time of extension [s]
Low–high	16 ± 2.9	70 ± 16.5 *	1.6 ± 0.2 *	2.1 ± 0.2 *
Parallel	14 ± 3.4	81 ± 17.2	0.99 ± 0.2	1.3 ± 0.1
Rotational	15 ± 1.6	80 ± 7.6	1.01 ± 0.1	1.3 ± 0.1
		**For the Trunk**		
	Min angle [°]	Max angle [°]	Time of forward movement [s]	Time of backward movement [s]
Low–high	−12 ± −2.0 *	30 ± 3.8 *	2.5 ± 0.3 *	1.2 ± 0.2 *
Parallel	−5 ± −0.6	36 ± 4.8	1.54 ± 0.2	0.7 ± 0.1
Rotational	−3 ± −0.1	41 ± 6.1	1.55 ± 0.2	0.7 ± 0.1

* a statistically significant different result.

## Data Availability

Available on request via email.

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
