# Peer review of "Description of Telemark Skiing Technique Using Full Body Inertial Measurement Unit"

_sensors, 2023, doi:10.3390/s23073448_

Round 1

Reviewer 1 Report

The authors proposed comparing three types of turns on telemark skis through a biomechanical description of each skiing technique.

  • Unfortunately, I believe this is not the proper journal for this paper. The authors have not presented any novelty in the scope of this journal. The data processing methods and the sensors used are well known. The authors just used them for a specific application. No evidence of novelty or contribution to the field could be found.
  • Even if it is a short communication, the work needs to present something novel, not just a new application in an area where the techniques have already been applied. Therefore, I recommend that authors add a few lines in section I - Introduction or section IV - Discussion to highlight why this paper presents some novelty in the field.

  • The authors could improve figure 3. The arrows are not indicating the phenomenon in the best way

  • The graph in Figure 2 has no labels for axes or units.

Author Response

Dear Reviewer
In the topics of the journal you can find the use of sensors in practice and I believe that our work is included in this topic. I sent this paper after initial editor approval as a "communication" type. So please rate this paper not as an original article but as communication. I realize that the method is not new and I am using a standard measurement set, but this is an example of the application of measurements. And with regard to the study of telemark skiing technique, I have not found similar reports.

Reviewer 2 Report

In this research, the authors investigated three types of turning techniques based on a biomechanical view using an inertial measurement unit (IMU). This research could benefit in preventing injury risk during telemark skiing. However, some improvements are required as follows:

Main questions:

1.           Please provide more information about the IMU equipment used in this research. Such as type, sensor number, and accuracy.

2.           What kind type of data is used for analysis? Is Euler angle or Quaternion? Also, because the rotational data were shown in tables, the definition of the joints needs to be given.

3.           The authors only discussed the knee, hip, and trunk joints. Are these three joints enough for analysis? Other important joints in the upper body, such as arm joints, are also important for analyzing the body's biomechanical movements.

4.           Figure 2 (3) in Line 176 has no labels on the x-axis and y-axis. Please add them to the figure. Moreover, the resolution of the figure needed to be higher. Please use vector images instead of pixel images if possible

Minor recommendation:

1.           Line 176: The figure numbers were wrong.

Reviewer 3 Report

Thank you for the possibility to review this manuscript. The topic is interesting and even if it is for a specific sector, I think it is for the scientific community. The manuscript is about the study of the telemark skiing technique through IMU sensors, this makes the study of interest, especially because it was highlighted how this methodology could decrease the risk of injuries. The methodology is well-structured but the discussion requires attention, it is poor of references and the topic “injury” is not anymore mentioned. Furthermore and most important, I think, before the publication of this study, the Ethic committee approval number should be written within the manuscript. 

Specific comments:

Line 76: please specify SD what is (standard deviation: SD)

Please, specify how the sample was involved; the eligibility criteria; if documents were signed; and if the principles of Helsinki were followed. Please, specify if the study was approved by an Ethic committee.

Please, implement the methods section with the study design paragraph.

Please, start the discussion highlighting the main findings of the study.

Please, try to improve the discussion with references. In this form is a narrative description of the results, there is no discussion of the findings.

Please, add in the discussion a section of the limits of the study and possible future application of the findings.

Please, add the conclusions of the study.

Round 2

Reviewer 1 Report

The authors addressed all my queries. I have no further comments.

Author Response

-

Reviewer 2 Report

The authors have replied to my comments. However, the quality of the figures could be improved. The discussion part contains many of the same sentences compared to other published papers, which can be detected by iThenticate.

Author Response

-

Round 3

Reviewer 2 Report

The plagiarism problems have been reduced.

Author Response

-